# ALPHAFOLD DISTILLATION FOR IMPROVED INVERSE PROTEIN FOLDING

## ABSTRACT

Inverse protein folding, i.e., designing sequences that fold into a given three-dimensional structure, is one of the fundamental design challenges in bio-engineering and drug discovery. Traditionally, inverse folding mainly involves learning from sequences that have an experimentally resolved structure. However, the known structures cover only a tiny space of the protein sequences, imposing limitations on the model learning. Recently proposed forward folding models, e.g., AlphaFold, offer unprecedented opportunity for accurate estimation of the structure given a protein sequence. Naturally, incorporating a forward folding model as a component of an inverse folding approach offers the potential of significantly improving the inverse folding, as the folding model can provide a feedback on any generated sequence in the form of the predicted protein structure or a structural confidence metric. However, at present, these forward folding models are still prohibitively slow to be a part of the model optimization loop during training. In this work, we propose to perform knowledge distillation on the folding model's confidence metrics, e.g., pTM or pLDDT scores, to obtain a smaller, faster and end-to-end differentiable distilled model, which then can be included as part of the structure consistency regularized inverse folding model training. Moreover, our regularization technique is general enough and can be applied in other design tasks, e.g., sequence-based protein infilling. Extensive experiments show a clear benefit of our method over the non-regularized baselines. E.g., in inverse folding design problems we observe up to 3% improvement in sequence recovery and up to 45% improvement in protein diversity, while still preserving structural consistency of the generated sequences.

## 1 INTRODUCTION

To date, 8 out of 10 top selling drugs are engineered proteins (Arnum, 2022). For functional protein design, it is often a pre-requisite that the designed protein folds into a specific three-dimensional structure.The fundamental task of designing novel amino acid sequences that will fold into the given 3D protein structure is named inverse protein folding. Inverse protein folding is therefore a central challenge in bio-engineering and drug discovery.

Computationally, inverse protein folding can be formulated as exploring the protein sequence landscape for a given protein backbone to find a combination of amino acids that supports a property (e.g. structural consistency). This task, computational protein design, has been traditionally handled by learning to optimize amino acid sequences against a physics-based scoring function (Kuhlman et al., 2003). In recent years, deep generative models have been proposed to solve this task, which consist of learning a mapping from protein structure to sequences (Jing et al., 2020; Cao et al., 2021; Wu et al., 2021; Karimi et al., 2020; Hsu et al., 2022; Fu & Sun, 2022). These approaches frequently use high amino acid recovery with respect to the ground truth sequence (corresponding to the input structure) as one success criterion. Other success criteria are high TM score (reflecting structural consistency) and low perplexity (measuring likelihood to the training/natural sequence distribution). However, such criteria solely ignore the practical purpose of inverse protein folding, i.e., to design novel and *diverse* sequences that fold to the desired structure and thus exhibit novel functions.

In parallel to machine learning advances in inverse folding, notable progresses have been made recently in protein representation learning (Rives et al., 2021; Zhang et al., 2022), protein structure

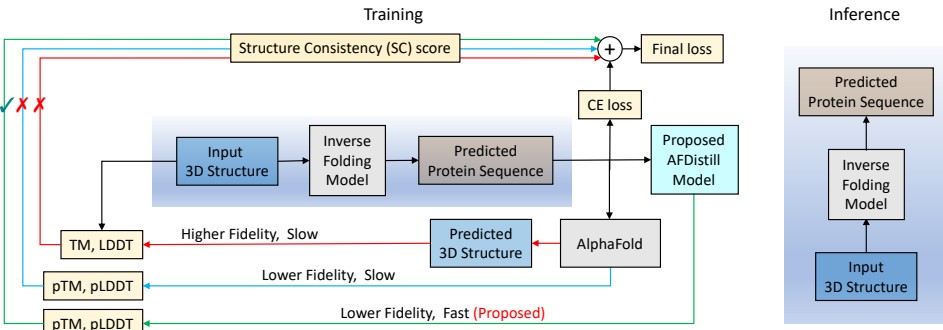

Figure 1: Overview of the proposed system. The traditional inverse protein folding (designing sequences that fold into a given 3D structures) is augmented by our proposed AFDistill model to increase the diversity of generated sequences while maintaining consistency with the given structure. One way of doing this (red line) would be to use forward protein folding models, e.g., AlphaFold, estimate structure from generated sequence, compare it with the ground truth to compute metrics such as TM or LDDT and, finally, regularize the original training loss (usually cross-entropy (CE)). However, inference through folding models is slow (see Fig. 2), making them impractical to be a part of the optimization loop. Alternatively, bypassing structure estimation, the folding model's internal confidence metrics, such as pTM or pLDDT can be used instead (blue line). This results in lower fidelity solutions that are still slow. Instead, in this work, we propose to distill the confidence metrics of AlphaFold into a smaller, faster, and differentiable model (referred as AFDistill). AFDistill is trained to maintain the comparable accuracy of the AlphaFold-estimated pTM/pLDDT, which now can be seamlessly used as part of the training loop (green line). The inference of the improved inverse folding model remains unmodified and is shown on the right side of the figure.

prediction from sequences (Jumper et al., 2021; Baek et al., 2021b), as well as in conditional protein sequence generation (Das et al., 2021; Anishchenko et al., 2021). These lines of works have largely benefited by learning from millions of available protein sequences (that may or may not have a resolved structure) in a self/un-supervised pre-training paradigm. Such large-scale pre-training has immensely improved the information content and task performance of the learned model. For example, it has been observed that structural and functional aspects emerge from a representation learned on broad protein sequence data (Rives et al., 2021). In contrast, inverse protein folding has mainly focused on learning from sequences that do have an experimentally resolved structure. Those reported structures cover only less than 0.1% of the known space of protein sequences, limiting the learning of the inverse folding model. In this direction, a recent work has trained an inverse folding model from scratch on millions of AlphaFold-predicted protein structures (in addition to tens of thousands of experimentally resolved structures) and shown performance improvement in terms of amino acid recovery (Hsu et al., 2022). However, such large-scale training from scratch is computationally expensive. A more efficient alternative would be to use the guidance of an already available forward folding model pre-trained on large-scale data in training the inverse folding model.

In this work we construct a framework where the inverse folding model is trained using a loss objective that consists of regular sequence reconstruction loss, augmented with an additional *structure consistency loss (SC)* (see Fig. 1 for the system overview). The straightforward way of implementing this would be to use forward protein folding models, e.g., AlphaFold, to estimate the structure from generated sequence, compare it with ground truth and compute TM score to regularize the training. However, a challenge in using Alphafold (or similar) directly is the computational cost associated with its inference(see Fig. 2), as well as the need of ground truth reference structure. Internal confidence structure metrics from the forward folding model can be used instead. However, that approach is still slow for the in-the-loop inverse folding model optimization. To address this, in our work we: **(i)** Perform *knowledge distillation* on AlphaFold and include the resulting model, AFDistill (frozen), as part of the regularized training of the inverse folding model (we term this loss structure consistency (SC) loss). The main properties of AFDistill model are that it is fast, accurate and end-to-end differentiable. **(ii)** Perform extensive evaluations, where the results on standard structure-guided sequence design benchmarks show that our proposed system outperforms existing baselines in terms of lower perplexity and higher amino acid recovery, while maintaining closeness to original protein

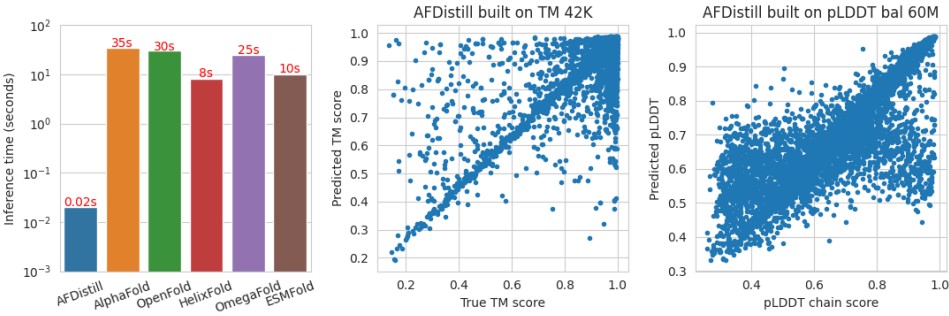

Figure 2: Average inference time for a protein sequence of length 500 by our AFDistill model versus the alternatives (left plot). Note that AFDistill fast performance holds for longer sequences: sequence length 1024 - inference time 0.028s, sequence length 2048 - inference time 0.035s. The timings for AlphaFold and OpenFold (Ahdritz et al., 2021) exclude the MSA searching time, which can vary from a few minutes to a few hours. Note that the values for HelixFold (Fang et al., 2022), OmegaFold (Wu et al., 2022) and ESMFold (Lin et al., 2022) are taken from their respective published results. As can be seen, the inference time of the current alternatives is too slow, which makes them impractical to be included as part of the model optimization loop. On the other hand, AFDistill is fast, accurate and end-to-end differentiable. The middle plot shows true and AFDistill predicted TM scores on TM distillation datasets (Pearson's correlation is 0.77) (see Section 3 for details). The right plot shows a similar scatter plot of the (averaged) true and the predicted pLDDT values on the pLDDT distillation dataset (Pearson's correlation is 0.76).

structure. More interestingly, we improve diversity in the designed sequences, one of the main goals of protein design. As a result of a trade-off between sequence recovery vs structure recovery, our regularized model yields better sequence diversity for a given structure, consistent with the fact that even small (35-40 amino acid) protein fold holds a 'sequence capacity' exceeding $10^{23}$ (Tian & Best, 2017). Note that our regularization technique is not limited to the inverse folding design and, as we show, can be applied to other applications, such as sequence-based protein infilling, where we also observe performance improvement over the baseline. **(iii)** Finally, the estimated structure consistency metric can either be used as part of the regularization of an inverse folding or infilling, during any other protein optimization tasks (e.g., (Moffat et al., 2021)) which would benefit from structural consistency estimation of the designed protein sequence, or as a cheap surrogate of AlphaFold that provides scoring of a given protein, reflecting its structural content.

## 2 RELATED WORK

**Forward Protein Folding.** Several powerful computational approaches have recently been proposed for the task of forward folding, namely predicting the structure of a protein given its sequence. In particular, AlphaFold (Jumper et al., 2021) makes use of multiple sequence alignments (MSAs) along with pairwise features. Building on similar ideas, RoseTTAFold (Baek et al., 2021a) is a three-track network combining information at the sequence level, the 2D distance map level, and the 3D coordinate level. On the implementation side, OpenFold (Ahdritz et al., 2021) is a PyTorch reproduction of Alphafold. However, MSAs of homologous proteins (e.g., antibodies, orphan proteins) are not always available and the MSA search adds significant overhead in inference time. Therefore, MSA-free approaches such as OmegaFold (Wu et al., 2022), HelixFold (Fang et al., 2022), and ESMFold (Lin et al., 2022) have been subsequently proposed. These architectures make use of large pretrained language models to produce predictions that are more accurate than AlphaFold and RoseTTAFold given a single sequence as input, and on par when given full MSAs as input.

**Inverse Protein Folding.** Several algorithms have also been proposed recently for the inverse protein folding problem, namely the task of identifying amino acid sequences that fold to a desired structure. (Norn et al., 2020) proposed a deep learning approach that optimizes over the entire folding landscape by backpropagating gradients through the trRosetta structure prediction network (Yang et al., 2020). (Anand et al., 2022) introduced a deep neural network architecture that explicitly models side-chain conformers in a structure-based context. Instead of making use of

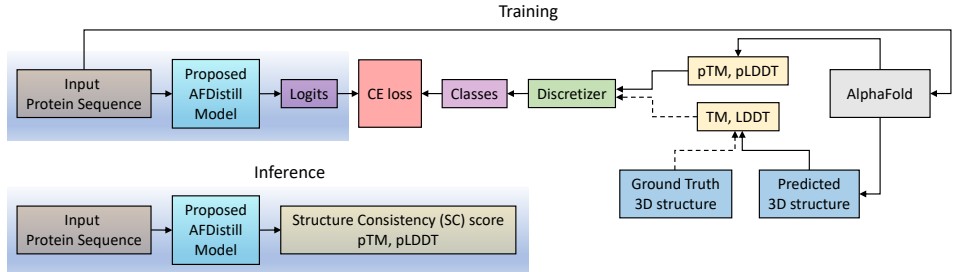

Figure 3: Distillation overview. Top diagram shows the training of AFDistill. The scores from AlphaFold's confidence estimation are denoted as pTM and pLDDT, while the scores which are computed using ground truth and the AlphaFold's predicted 3D structures are denoted as TM and LDDT. These values are then discretized and treated as class labels during cross-entropy (CE) training. Note that the training based on TM/LDTT is limited since the number of known ground truth structures is small. The bottom diagram shows the inference stage of AFDistill, where for each protein sequence it estimates pTM and pLDDT scores.

trRosetta, (Jendrusch et al., 2021) presented a framework that incorporates the structure prediction algorithm AlphaFold (Jumper et al., 2021) into an optimization loop to generate protein sequences. However, as noted above, the inference of AlphaFold is expensive due to MSA search, hence employing AlphaFold in an optimization loop for protein sequence generation is very cumbersome. Though faster, the inference of MSA-free approaches (e.g. OmegaFold, HelixFold and EMSFold) is still too slow for their use in an optimization loop.

In this work, we propose knowledge distillation from the forward folding algorithm AlphaFold, and build a student model that is small, practical and accurate enough. We show that the distilled model can be efficiently used within the inverse folding model optimization loop and improve quality of designed protein sequences.

## 3 ALPHAFOLD DISTILL

Knowledge distillation (Hinton et al., 2015) transfers knowledge from a large complex model, in our case AlphaFold, to a smaller one, here this is the AFDistill model (see Fig. 3). Traditionally, the distillation would be done using soft labels, which are probabilities from AlphaFold model, and hard labels, the ground truth classes. However, in our case we do not use the probabilities as they are harder to collect or unavailable, but rather the model's predictions (pTM/pLDDT) and the hard labels, TM/LDDT scores, computed based on AlphaFold's predicted 3D structures.

### 3.1 SCORES TO DISTILL

*TM-score* (Template Modeling score) (Zhang & Skolnick, 2004), is the mean distance between structurally aligned $C_\alpha$ atoms scaled by a length-dependent distance parameter. *LDDT* (Local Distance Difference Test) (Mariani et al., 2013) is the average of four fractions computed using distances between all pairs of atoms based on four tolerance thresholds (0.5Å, 1Å, 2Å, 4Å) within 15Å inclusion radius. The range of both metrics is (0,1), and the higher values represent more similar structures.

*pTM* and *pLDDT* are the AlphaFold-predicted metrics for a given input protein sequence, corresponding to the reconstructed 3D protein structure, which represent model's confidence of the estimated structure. *pLDDT* is a local per-residue score (pLDDT chain is another score that simply averages per-residue pLDDTs across the chain), while *pTM* is a global confidence metric for assessing the overall chain reconstruction. In this work we interpret these metrics as the quality or validity of the sequence for the purpose of downstream applications (see Section 4).

### 3.2 DATA

Using Release 3 (January 2022) of AlphaFold Protein Structure Database (Varadi et al., 2021), we collected a set of 907,578 predicted structures. Each of these predicted structures contains 3D

| Release 3 (January 2022) | | Release 4 (July 2022) | |
|---|---|---|---|
| Name | Size | Name | Size |
| Original | 907,578 | Original | 214,687,406 |
| TM 42K | 42,605 | pLDDT balanced 1M | 1,000,000 |
| TM augmented 86K | 86,811 | pLDDT balanced 10M | 10,000,000 |
| pTM synthetic 1M | 1,244,788 | pLDDT balanced 60M | 66,736,124 |
| LDDT 42K | 42,605 | | |
| pLDDT 1M | 905,850 | | |

Table 1: Statistics from January 2022 (left side) and July 2022 (right size) releases of the AlphaFold database. For the earlier release, we created multiple datasets for pTM and pLDDT estimation, while for the later, larger release we curated datasets only for pLDDT estimation.

coordinates of all the residue atoms as well as the per-resiude pLDDT confidence scores. To avoid any data leakage to the downstream aplications, we first filtered out the structures that are part of the validation and test splits of CATH 4.2 dataset (discussed in Section 4). Then, using the remaining structures, we created our pLDDT 1M dataset (see Table 1), where each protein sequence is paired with the sequence of per-residue pLDDTs. We also truncated proteins up to length 500 to reduce computational complexity of AFDistill training.

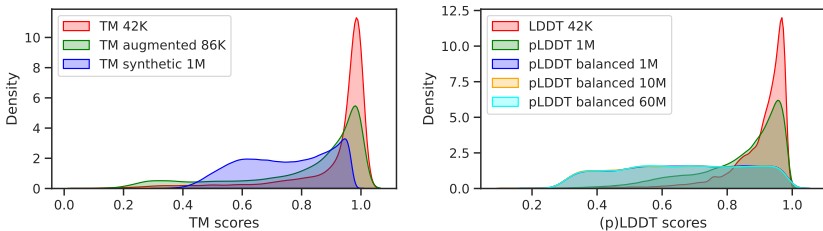

Figure 4: Distribution of the (p)TM/(p)LDDT scores in various datasets used in AFDistill training.

We also created datasets which are based on the true TM and LDDT values using the predicted AlphaFold structures. Specifically, using the PDB-to-UniProt mapping, we selected a subset of samples with matching ground truth PDB sequences and 3D structures, resulting in 42,605 structures. We denote these datasets as TM 42K and LDDT 42K (see Table 1). In Fig. 4 we show the score density distribution of each dataset. As can be seen, the TM 42K and LDDT 42K are very skewed to the upper range of the values. To mitigate this data imbalance, we curated two additional TM-based datasets. TM augmented 86K was obtained by augmenting TM 42K with a set of perturbed original protein sequences, estimating their structures with AlphaFold, computing corresponding TM-score, and keeping the low and medium range TM values. pTM synthetic 1M was obtained by generating random synthetic protein sequences and feeding them to AFDistill (pre-trained on TM 42K data), to generate additional data samples and collect lower-range pTM values. The distribution of the scores for these additional datasets is also shown in Fig. 4, where both TM augmented 86K and pTM synthetic 1M datasets are less skewed, covering lower (p)TM values better.

Finally, using Release 4 (July 2022), containing over 214M predicted structures, we plotted its distribution density of pLDDT values and observed similar high skewness towards upper range. To fix this, we rebalanced the data by filtering out samples with upper-range mean-pLDDT values (also called pLDDT chain). The resulting dataset contains 60M sequences, for which we additionally created 10M and 1M versions, see Fig. 4 for their density.

## 3.3 MODEL

AFDistill model is based on ProtBert (Elnaggar et al., 2020), a Transformer BERT model (420M parameters) pretrained on a large corpus of protein sequences using masked language modeling. For our task we modify ProtBert head by setting the vocabulary size to 50, corresponding to discretizing pTM/pLDDT in range (0,1). For pTM (scalar) the output corresponds to the first ⟨CLS⟩ token of the output sequence, while for pLDDT (sequence) the predictions are made for each residue position.

| Training data | Validation CE loss | Training data | Validation CE loss |
|---|---|---|---|
| TM 42K | **1.10** | LDDT 42K | 3.39 |
| TM augmented 86K | 2.12 | pLDDT 1M | 3.24 |
| pTM synthetic 1M | 2.55 | pLDDT balanced 1M | 2.63 |
| | | pLDDT balanced 10M | 2.43 |
| | | pLDDT balanced 60M | **2.40** |

Table 2: Validation loss of AFDistill on datasets from Table 1 (For more details, see Tables 3 and 4.

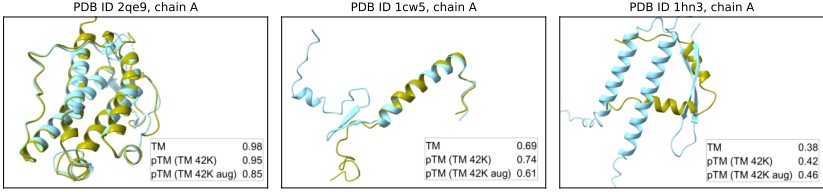

Figure 5: Examples of 3D protein structures from the dataset, corresponding to high, medium, and low actual TM scores (top row in legend), as well as AFDistill predictions, trained on TM 42K (middle row) and TM augmented 86K (bottom row).

## 3.4 DISTILLATION EXPERIMENTAL RESULTS

In this section we present evaluation results of the model after training it on the presented datasets. We note that to further improve the data imbalance problem, during training we employed weighted sampling in the minibatch generation as well as used Focal loss (Lin et al., 2017) in place of the traditional cross-entropy loss. The results for (p)TM-based datasets are shown in Table 2. We see that AFDistill trained on TM 42K dataset performed the best, followed by the dataset with the augmentations, and the synthetic data. For the (p)LDDT-based datasets, we observe that increasing the scale, coupled with the data balancing, improves the validation performance. In Fig. 2 we show scatter plots of the true vs pTM scores and pLDDT values on the entire validation set. We see a clear diagonal pattern in both plots, where the predicted and true values match. There are also some number of incorrect predictions (reflected along the off-diagonal), where we see that for the true scores in the upper range, the predicted scores are lower, indicating that AFDistill tends to underestimate them. Finally, in Fig. 5 and 6, we show a few examples of the data samples together with the corresponding AFDistill predictions. Fig. 2 and Fig. 12, 13 (in Appendix) also show plots of SC (pTM or pLDDT) versus TM score, indicating that AFDistill is a viable tool for regularizing protein representation to enforce structural consistency or structural scoring of protein sequences, reflecting its overall composition and naturalness (in terms of plausible folded structure).

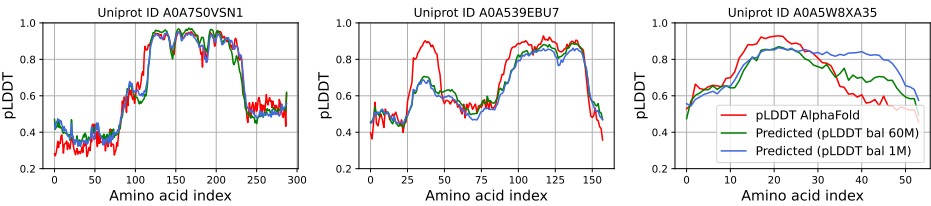

Figure 6: Dataset examples of the per-residue predictions for two AFDistill models (blue and green lines), build on pLDDT balanced 1M and 60M datasets, versus the AlphaFold predictions (red line).

## 4 INVERSE PROTEIN FOLDING DESIGN

In this section we demonstrate the benefit of applying AFDistill as a structure consistency (SC) score for solving the task of inverse protein folding, as well as for the protein sequence infilling as a means to novel antibody generation. The overall framework is presented in Fig. 1 (following the green line in the diagram), where the traditional inverse folding model is regularized by our SC

score. Specifically, during training, the generated protein is fed into AFDistill, for which it predicts pTM or pLDDT score, and combined with the original CE training objective results in

$$\mathcal{L} = \mathcal{L}_{CE} + \alpha\mathcal{L}_{SC}, \tag{1}$$

where $\mathcal{L}_{CE} = \sum_1^N \mathcal{L}_{CE}(\mathbf{s}_i, \hat{\mathbf{s}}_i)$ is the CE loss, $\mathbf{s}_i$ is the ground truth and $\hat{\mathbf{s}}_i$ is the generated protein sequence, $\mathcal{L}_{SC} = \sum_{i=1}^N (1 - SC(\hat{\mathbf{s}}_i))$ is the structure consistency loss, $N$ is the number of training sequences, and $\alpha$ is the weighting scalar for the SC loss, in our experiment it is set to 1.

## 4.1 METRICS

To measure the quality of the prediction designs, we compute the following set of sequence evaluation metrics. *Recovery* (range (0, 100) higher is better) is the average number of exact matches between predicted and ground truth sequences, normalized by the length of the alignment. *Diversity* (range (0, 100) higher is better) of a predicted protein set is the complement of the average recovery computed for all pairwise comparisons in the set. While in general the recovery and diversity tend to be inversely correlated, i.e., higher recovery leads to lower diversity, and vice versa, we are interested in models that achieve high recovery rates and be able to maintain high protein sequence diversity. *Perplexity* measures the likelihood of a given sequence, lower values mean better performance. Finally, for structure evaluation, we use *TM-score* as well as the *structure consistency (SC)* score, which is the AFDistill's output (pTM/pLDDT) for a given input.

## 4.2 RESULTS

We present experimental results for several recently proposed deep generative models for protein sequence design accounting for 3D structural constraints. For the inverse folding tasks we use CATH 4.2 dataset, curated by (Ingraham et al., 2019). The training, validation, and test sets have 18204, 608, and 1120 structures, respectively. While for protein infilling we used SabDab (Dunbar et al., 2013) dataset curated by (Jin et al., 2021) and focus on infilling CDR-H3 loop. The dataset has 3896 training, 403 validation and 437 test sequences.

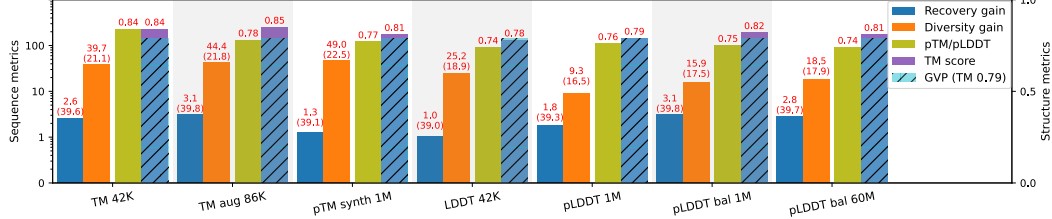

Figure 7: Evaluation results of GVP trained with SC regularization. The horizontal x-axis shows various datasets for AFDistill pretraining, the left vertical y-axis shows sequence metrics (recovery and diversity gains), while the right y-axis shows structure metrics (TM and SC scores). Blue and orange solid bars show recovery and diversity gains (top number - percentage, bottom - actual value) of SC-regularized GVP over the vanilla GVP baseline. Solid olive bar shows the predicted SC (pTM or pLDDT, depending on AFDistill model), while the purple bar is the test set TM score (structures predicted by AlphaFold). Overlaid dashed cyan bar is the TM score of the baseline GVP. We can see that overall TM 42K and TM augmented 86K pretrained AFDistill achieve the best overall performance, with high diversity and moderate improvement in sequence and structure recovery.

**GVP** Geometric Vector Perceptron GNNs (GVP) (Jing et al., 2020) is the inverse folding model, that for a given target backbone structure, represented as a graph over the residues, replaces dense layers in a GNN by simpler layers, called GVP layers, directly leveraging both scalar and geometric features. This allows for the embedding of geometric information at nodes and edges without reducing such information to scalars that may not fully capture complex geometry. The results of augmenting GVP training with SC score regularization are shown in Fig. 7 (see also Appendix, Table 8 for additional results). Baseline GVP with no regularization achieves 38.6 in recovery, 15.1 in diversity and 0.79 in TM score on the test set. It can be seen that there is a consistent improvement in sequence recovery gain (1-3%) over the original GVP and significant diversity gain (up to

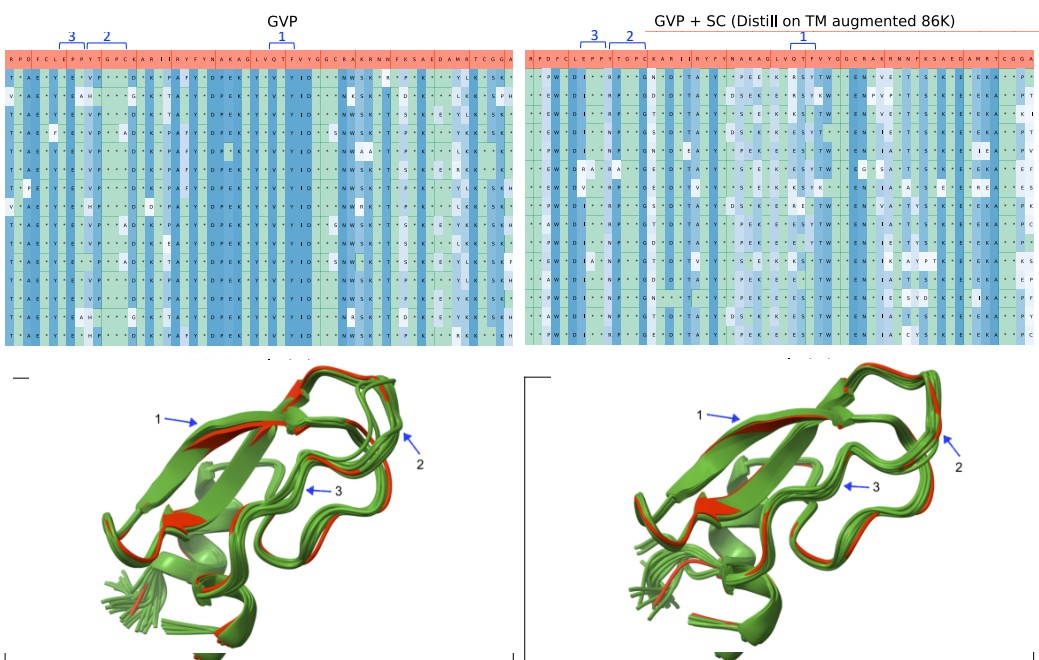

Figure 8: Comparison between baseline GVP (left) and SC regularized GVP (right), where AFDistill was pre-trained on TM augmented 86K dataset. The tables display 15 generated protein sequences from each model, top red row is the ground truth. Green cell with the * indicates amino acid identity compared to ground truth (top red row), while blue cell shows novelty. The shade of blue cell represents the frequency of the amino acid in that column (darker → more frequent, lighter → rare). Therefore, the method with high recovery and diversity rates will have many green and light blue cells. Bottom plots show AlphaFold estimated structures (green) and the ground truth (red). SC-regularized GVP, while having high sequence diversity, still results in accurate reconstructions, while GVP alone has more inconsistencies, marked with arrows.

45%) of the generated protein sequences, when we employ SC regularization. At the same time the estimated structure (using AlphaFold) remains close to the original as measured by the high TM score. We also observed that pTM-based SC scores had overall better influence on the model performance as compared to pLDDT-based ones. It should be further noted, that the validation performance of AFDistill on the distillation data is not always reflective of the performance on the downstream applications, as AFDistill trained on TM augmented 86K overall performs better than TM 42K, while having slightly worse validation CE loss (Table 2). This observation indicates that the augmented models might be less biased by the teacher model, hence enables more generalized representation learning of sequence–structure relationship and provides more performance boost to the inverse folding model.

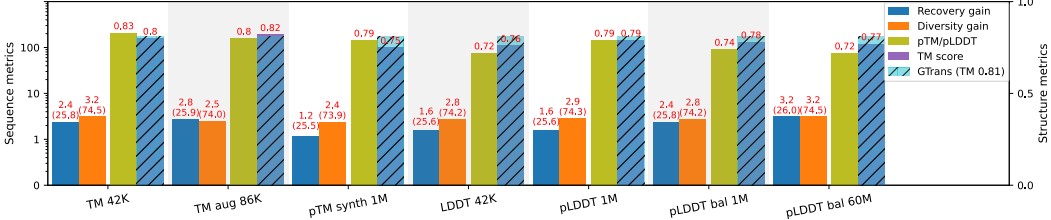

Figure 9: Evaluation results of Graph Transformer model trained with SC score regularization.

To further illustrate the effect of recovery and diversity, we show in Fig. 8 protein sequences and AlphaFold-generated 3D structures of GVP and GVP+SC models, where the latter model achieves higher diversity of the sequence while retaining accurate structure of the original protein. Here, the

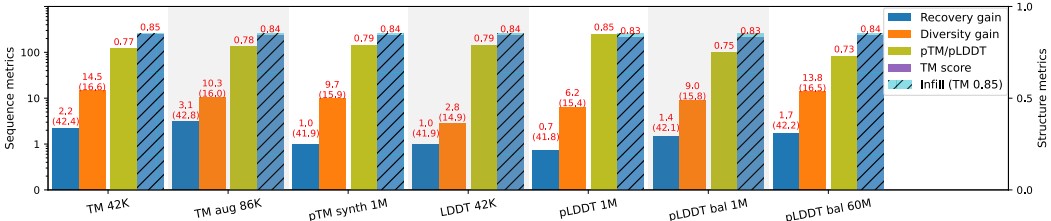

Figure 10: Evaluation results of protein infilling model trained with SC regularization.

recovery is 40.8 and diversity is 11.2 for GVP, while for GVP+SC it is 42.8 and 22.6, respectively, confirming that GVP+SC achieves higher recovery and diversity. The bottom plots show AlphaFold estimated structures (green) and the ground truth (red). It can be seen that for GVP+SC, the high sequence diversity still results in very accurate reconstructions (for this example, average TM score is 0.95), while GVP alone shows more inconsistencies, marked with blue arrows, (TM score 0.92).

**Graph Transformer** We evaluated the effect of SC score on Graph Transformer (Wu et al., 2021), another inverse folding model, which seeks to improve standard GNNs to represent the protein 3D structure. Graph Transformer applies a permutation-invariant transformer module after GNN module to better represent the long-range pair-wise interactions between the graph nodes. The results of augmenting Graph Transformer training with SC score regularization are shown in Fig. 9 (see also Appendix, Table 9 for additional results). Baseline model with no regularization has 25.2 in recovery, 72.2 in diversity and 0.81 in TM score on the test set. As compared to GVP (Fig. 7), we can see that for this model, the recovery and diversity gains upon SC regularization are smaller. We also see that TM score of regularized model (TM 42K and TM augmented 86K pretraining) is slightly higher as compared to pLDDT-based models.

**Protein Infilling** Our proposed structure consistency regularization is quite general and not limited to the inverse folding task. Here we show its application on protein infilling task. Recall, that while the inverse folding task considers generating the entire protein sequence, conditioned on a given structure, infilling focuses on filling specific regions of a protein conditioned on a sequence/structure template. The complementarity-determining regions (CDRs) of an antibody protein are of particular interest as they determine the antigen binding affinity and specificity. We follow the framework of (Jin et al., 2021) which formulates the problem as generation of the CDRs conditioned on a fixed framework region. We focus on CDR-H3 and use a baseline pretrained protein model ProtBERT (Elnaggar et al., 2020) finetuned on the infilling dataset, and use ProtBERT+SC as an alternative (finetuned with SC regularization). The CDR-H3 is masked and the objective is to reconstruct it using the rest of the protein sequence as a template. The results are shown in Fig. 10 (see also Appendix, Table 10 for additional results). Baseline model achieves 41.5 in recovery, 14.5 in diversity, and 0.80 in TM score on the test set. Similar as for the other applications, we see an improvement in the sequence recovery and even bigger gain in diversity, while using the AFDistill pretrained on TM 42K and TM augmented 86K, together with the pLDDT balanced datasets. TM score shows that the resulting 3D structure remains close to the original, confirming the benefit of using SC for training regularization.

## 5 CONCLUSION

In this work we introduce AFDistill, a distillation model based on AlphaFold, which for a given protein sequence estimates its structural consistency (SC: pTM or pLDDT) score. We provide experimental results to showcase the efficiency and efficacy of the AFDistill model in high-quality protein sequence design, when used together with a graph neural net based inverse folding model or large protein language model for sequence infilling. Our AFDistill model is small and accurate enough so that it can be efficiently used for regularizing structural consistency in protein optimization tasks, maintaining sequence and structural integrity, while introducing diversity and variability in the generated proteins.

## REPRODUCIBILITY STATEMENT

For the sake of reproducibility, we use publicly available datasets and describe the data in detail in Section 3.2. We provide training details in the Appendix. The code will be publicly released upon acceptance.

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

## A    BACKGROUND ON PROTEIN DESIGN

A protein is a linear chain of variable length made up of twenty amino acids, also called residues. These are denoted by 20 characters (A-Alanine, G-Glycine, I-Isoleucine, L-Leucine, P-Proline, V-Valine, F-Phenylalanine, W-Tryphtophan, Y-Tyrosine, D-Aspartic Acid, E-Glutamic Acid, R-Arginine, H-Histidine, K-Lysine, S-Serine, T-Threonine, C-Cystene, M-Methionine, N-Asparagine, Q-Glutamine). Each amino acid has the same core structure (backbone), consisting of alpha carbon atom $C_\alpha$, connected to an amino group $NH2$, carboxyl group $COOH$ and hydrogen atom $H$. The backbone is identical in all amino acids, while the variable group, called side chain, which is also attached to alpha carbon $C_\alpha$, is always different and determines the amino acid, including its chemical and mechanical properties. Amino acids are attached to each other by a covalent bond, known as peptide bond (carboxyl group $COOH$ of one amino acid and the amino group $NH2$ of the other amino acid combine, releasing water molecule $H2O$ and create a peptide bond). In this work, as is commonly done, we define protein 3D structure specified only by the $C_\alpha$ atoms of amino acids.

The protein inverse folding task is to draw a sequence from the true distribution of $n$-length sequences of amino acids $Y \in \{1, \ldots, 20\}$, conditioned on a fixed protein structure, such that the designed protein folds into that structure. The protein structure can be represented as an attributed graph $G = (V, E)$ with node features $V = \{v_1, \ldots, v_N\}$, describing each residue and edge features $E = \{e_{ij}\}$, capturing relationships between them. Thus, the final conditional distribution we are interested in modeling is: $P(Y|X) = p(y_i, \ldots, y_n|X)$, which is known as computational protein design task.

Protein structures are intrinsically dynamic and each structure thus possess high designability, i.e. the total number of amino acid sequences that can fold to a target protein structure is high, without losing stability of the structure. The highly designable structures always enjoy beneficial properties such as higher thermodynamic stability, mutational stability, fast folding, functional robustness, etc. Therefore, we need to learn a "soft" function that can model this high designability associated with a protein structure, i.e. generating diverse sequences for a given protein structure.

## B    ALPHAFOLD MODEL OVERVIEW

A schematic overview of AlphaFold model is shown in Fig. 11, which it takes as input a protein sequence and produces as output, among others, the predicted 3D structure, as well as the confidence estimates of its prediction, pTM and pLDDT, which measure the estimated confidence of how well the predicted and ground truth structures match.

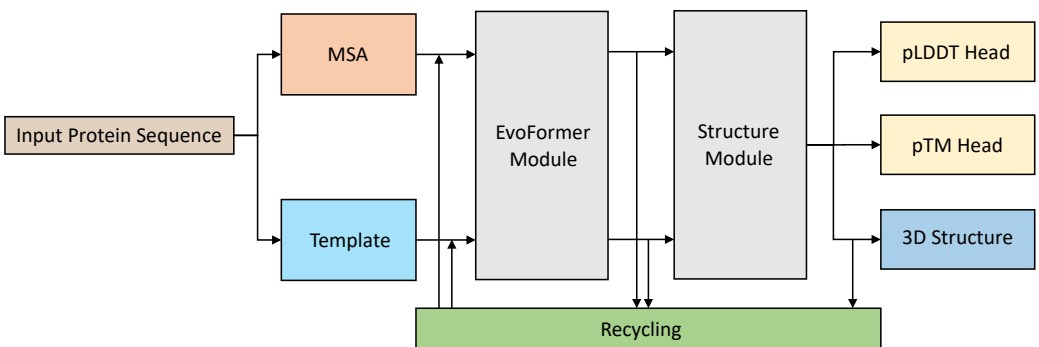

Figure 11: Overview of the inference stage in AlphaFold model. Given an input protein sequence, first the search is performed in genetic database to find similar sequences and construct multiple sequence alignments (MSA). Then a structure database search is done to find similar 3D structures and construct templates. The MSA and templates are fed into EvoFormer module, whose output is then sent to the Structure module, which is finally completed with the multiple output heads. The 3D structure head generates predicted 3D protein structure, while pLDDT and pTM heads estimate the confidence of the computed structure. Optionally, the generated structure together with the intermediate states are recycled and sent back to update/correct MSA and template representations for further processing and improvement.

## C  AFDISTILL TRAINING

Tables 3, 4 show the validation performance of AFDistill trained on each of the (p)TM-based and (p)LDDT-based datasets, respectively. Table 5 shows results on (p)LDDT chain-based datasets. Note that (p)LDDT chain is the dataset, similar to (p)TM datasets, where for each sequence we associate a single scalar, in this case the average of all the per-residue (p)LDDT values.

| Data | Training | | Validation |
| --- | --- | --- | --- |
| | Weighted sampling | Focal loss ($\gamma$) | CE loss |
| TM 42K | − | − | 1.33 |
| | + | − | 1.37 |
| | + | 1.0 | 1.16 |
| | + | 3.0 | **1.10** |
| | + | 10.0 | 1.29 |
| TM augmented 86K | − | − | **2.12** |
| | + | 1.0 | 2.15 |
| | + | 3.0 | 2.19 |
| | + | 10.0 | 2.25 |
| pTM synthetic 1M | − | − | 2.90 |
| | + | 1.0 | 2.75 |
| | + | 3.0 | **2.55** |
| | + | 10.0 | 3.20 |

Table 3: Validation CE loss for the AFDistill model trained on each of the (p)TM-based datasets. To address data imbalance during training, we employed weighted sampling for minibatch generation to so that the TM-scores cover their range (0,1) close to uniform distribution. Moreover, we also used Focal loss (Lin et al., 2017) in place of the standard cross-entropy (CE) loss (the evaluation is still done using CE loss across all the training setups). Based on the validation loss, we see that the AFDistill model trained on TM 42K dataset performed the best, followed by the dataset with augmentations, and the synthetic performed the worst. We also see that weighted sampling and focal loss do help in addressing the data imbalance problem, although for TM augmented 86K, the balanced augmentation seemed to help better and the best performance was for the case when no weighted sampling is applied and the traditional CE loss is used. As shown in Section 4, the validation performance on the distillation data may not always indicate the performance on the downstream applications, where in particular we observed that the Distill model, trained on TM augmented 86K dataset, overall performed better than TM 42K, while having slightly worse validation CE loss.

| Data | Training | | Validation |
| --- | --- | --- | --- |
| | Weighted sampling | Focal loss ($\gamma$) | CE loss |
| LDDT 42K | - | - | 3.47 |
| | + | 1.0 | 3.44 |
| | + | 3.0 | 3.42 |
| | + | 10.0 | **3.39** |
| pLDDT 1M | - | - | 3.27 |
| | + | 1.0 | 3.28 |
| | + | 3.0 | **3.25** |
| | + | 10.0 | 3.24 |

Table 4: Validation CE loss for AFDistill trained on each of the (p)LDDT-based datasets. We see that weighted sampling coupled with Focal loss, performed the best.

| Data | Training | | Validation |
|---|---|---|---|
| | Weighted Sampling | Focal loss ($\gamma$) | CE loss |
| LDDT chain 42K | - | - | 3.69 |
| | + | 1.0 | 3.57 |
| | + | 3.0 | 3.63 |
| | + | 10.0 | **3.59** |
| pLDDT chain 1M | - | - | 3.29 |
| | + | 1.0 | 3.36 |
| | + | 3.0 | **3.30** |
| | + | 10.0 | 3.31 |
| pLDDT chain balanced 1M | – | – | 2.45 |
| pLDDT chain balanced 10M | – | – | 2.24 |
| pLDDT chain balanced 60M | – | – | **2.21** |

Table 5: Validation CE loss for the AFDistill model trained on each of the (p)LDDT chain-based datasets. (p)LDDT chain is the dataset, similar to (p)TM datasets, where for each sequence we associate a single scalar, in this case the average of all the per-reside (p)LDDT values. Similar as before, we see that the use of weighted sampling coupled with Focal loss helps in boosting the model performance. We also see that increasing the scale of data (which is already balanced) improves the performance even further.

# D  GVP TRAINING DETAILS

An example of GVP training progress regularized by the structure consistency (SC) score computed by the AFDistill model (pre-trained on various (p)TM-based datasets) is shown in Fig. 12. This figure shows that although SC score may be less accurate on the absolute scale, on the relative scale we can see it accurately detecting decays and improvements in the sequence quality as the GVP trains. Similarly, in Fig. 13 we show scatter plots of estimated pTM versus true TM score for GVP-generated protein sequences regularized by SC score.

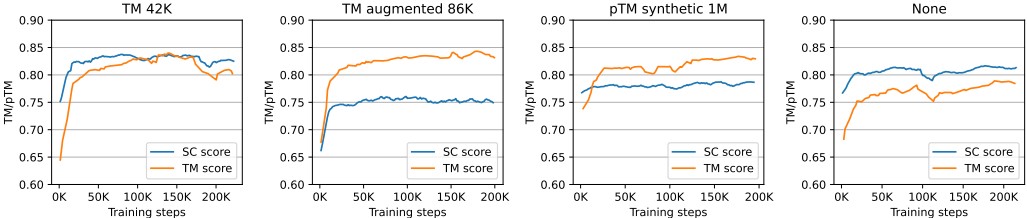

Figure 12: Example of the training progress (on CATH 4.2 dataset) of the GVP model regularized by the structure consistency (SC) score computed by the AFDistill model pre-trained on different datasets. Each plot shows the results for one of the Distill pre-training datasets, where the blue line represents the SC score computed by the AFDistill model (in this case generating pTM value), while the orange line shows the actual TM score computed between the ground truth structure and the AlphaFold's estimated 3D structures for a GVP-generated protein sequences. The last plot on the right shows the original, unregularized GVP training, where SC score was computed but never applied as part of the loss. It can be seen that SC correlates well with the TM score for TM 42K, while for others (TM augmented 86K and pTM synthetic 1M datasets) it tends to underestimate true TM score. Therefore, SC score may be less accurate on the absolute scale, while on the relative scale we can see that it can accurately detect decays and improvements in the sequence quality as the GVP trains. And the latter is of particular importance for SC to be a regularization loss during training, since it can clearly identify the ill-generated protein sequences early in the training (lower SC scores) and recognize well-defined sequences later during the training (higher SC scores).

## D.1  EFFECT OF USING AFDISTILL TRAINED FROM SCRATCH

We also experimented with AFDistill models trained from scratch (as opposed to starting from pre-trained ProtBert), but observed worse performance. As an example, we trained AFDistill from scratch on TM42K dataset. The validation CE loss during distillation was 1.5 (versus 1.1 when using pre-trained ProtBert model). Moreover, training of AFDistill model from scratch takes longer (3 days vs 1 day). When regularizing GVP with AFDistill from scratch, we get similar recovery rate (39.4 vs 39.6) but lower sequence diversity (15.9 vs 21.1), which confirms the benefit of common practice of fine-tuning the pretrained models as opposed to starting from random models weights.

## D.2  EFFECT OF STRUCTURE CONSISTENCY (SC) SCORE ON GVP PERFORMANCE

For protein design (e.g., using GVP as a base model) the objective is CE + SC (cross-entropy + AFDistill structure consistency score). In Fig. 14 we present the effect of SC magnitude on the GVP performance on the test set of CATH dataset. As can be seen, when only the CE term is present (the blue left most bar in both panels, representing the original GVP), the model is encouraged to recover the specific ground truth protein sequence for a given 3D structure, and this promotes model accuracy, and high amino acid recovery rate, while also resulting in low diversity. On the other hand, when only the SC term is present (the pink right most bar, reprenting CE+32*SC, i.e., when SC completely dominates and CE can be ignored), this results in poor and degenerated protein sequences. This is expected, since AFDistill alone cannot guide GVP which sequence it should generate to match the given input 3D structure. Recall, that AFDistill has no information about the structure, and since many of the relevant protein sequences can have high pTM/pLDDT, all of them could be good candidates, and this promotes high diversity and low recovery. Consequently, when both CE and SC terms are present and when appropriate balance between them is found (in our case

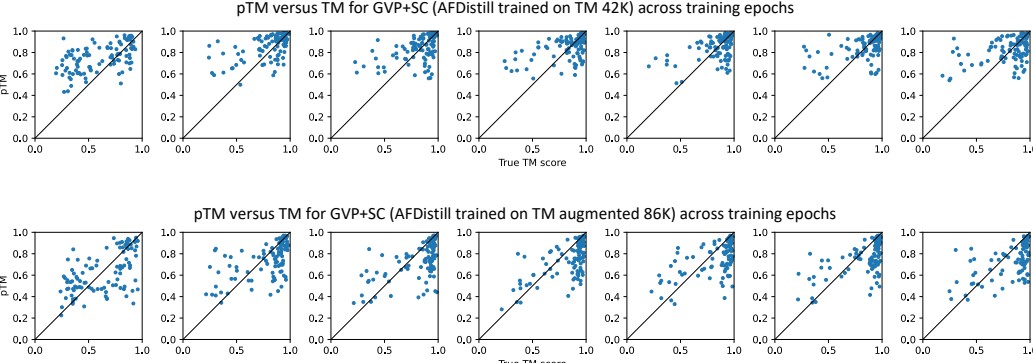

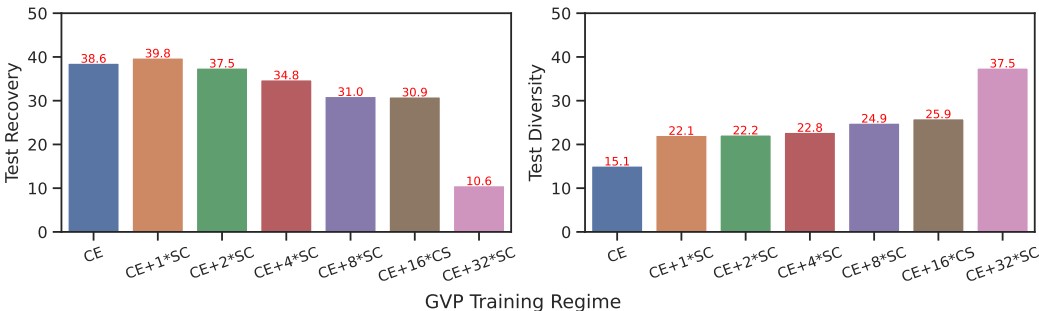

Figure 13: Estimated pTM versus true TM score (based on AlphaFold structure prediction) for GVP-generated protein sequences regularized by SC score. The top row shows results for SC computed by AFDistill model trained on TM 42K, while the bottom row is for AFDistill trained on TM augmented 86K. The columns in each row correspond to the progress as GVP trains. Note that the top row corresponds to the first left plot in Fig. 12, while the bottom row corresponds to the second plot in Fig. 12. It can be observed that in the earlier stages of GVP training, the generated protein sequences are of poor quality, reflected in pTM and TM scores that are spread across the (0,1) range. On the other hand, as the training progresses, the generated sequences are getting better and the pTM/TM score is concentrated more in the upper range. Another observation is that for AFDistill trained on TM 42K dataset, the predicted and true TM score are better aligned across the diagonal (compare with orange and blue lines on the left plot in Fig. 12), while for AFDistill trained on TM augmented 86K dataset, pTM tends to underestimate true TM score. These plots show that AFDistill is viable sequence scoring tool, which fairly accurately measures the structural consistency of the generated protein sequences. Combined with the fact that it is fast and end-to-end differentiable, shows its potential for many of the protein optimization problems.

Figure 14: The effect of Structure Consistency (SC) loss on the performance of GVP. Left panel shows the amino acid recovery rate and the right panel shows the diveristy rate on the test set of CATH dataset. The horizontal y-axis shows the different choices of objective function during training: CE is the cross-entropy loss, SC is the Structure Consistency score computed by AFDistill.

it is CE+SC, corresponding to the orange bar in both panels), we get a full benefit, i.e., the accurate recovery and high diversity of the generated protein sequences.

# E    ADDITIONAL PERFORMANCE COMPARISONS ON GVP

## E.1    ESM-IF

In this Section we compare GVP performance under different training scenarios and present the results in Table 6. The first row is the recovery rate of the original GVP model as reported in Jing et al. (2020). The following three rows are the results presented in the work of Hsu et al. (2022) (ESM-IF). Their evaluation showed that the vanilla GVP achieved a slightly higher recovery rate of 42.2. GVP+AlphaFold2 represents the GVP trained on augmented dataset (CATH + AlphaFold2-generated structure/sequence pairs). Interestingly, this simple data augmentation baseline showed worse performance as compared to the original GVP, and the authors had to significantly increase GVP capacity (from 1M to 21M)to get any benefit from the data augmentation. Moreover, note that such a data augmentation idea can also serve as the baseline for our approach of AFDistill regularization, since AFDistill was trained on AlphaFold2-generated data and it can be thought of as a compressed representation of that data.

The last two rows show our evaluation results of the vanilla GVP, achieving slightly lower base recovery rate of 38.6, while this same GVP but trained with AFDistill regularization achieves a small boost in recovery (39.6) and significant increase in the sequence diversity (as we showed in Fig. 7).

Therefore, comparing data augmentation and model distillation for the task of protein design, we see that for the original GVP model, AFDistill offers a clear advantage (compare third and last rows in Table 6), providing a modest boost in recovery, while significantly increasing diversity of the generated sequences. Moreover, the distillation overhead is amortized, as we train AFDistill once and use it in many downstream applications. The data augmentation would require additional computational cost in every downstream application.

On the other hand, our AFDistill regularization was able to improve the original GVP (with modest recovery and significant diversity gains). Moreover, the distillation overhead is amortized - train once use everywhere. We paid extra cost to train AFDistill but then applied it in many downstream tasks with little overhead. Data augmentation would require additional significant cost in every application. Finally, AFDistill offers unique advantage - it maintains recovery and structural consistency of the original model while introducing diversity into the generated sequences.

## E.2    PROTEINMPNN

In this section we present additional experimental results on ProteinMPNN model Dauparas et al. (2022). We used the publicly available code repository and data from https://github.com/dauparas/ProteinMPNN. We compared the results of original unmodified training of ProteinMPNN to the training using SC regularization (AFDistill model trained on TM aug 86K dataset). We evaluated both training regimes on amino acid recovery, protein sequence diversity and perplexity. We also varied ProteinMPNN internal parameter, which adds noise to the input backbone protein structure. Table 7 presents the results. We can see that Structure Consistency (SC) regularization maintains recovery and perplexity rates while improving the diversity of the generated protein sequences. Backbone noise, which is a part of ProteinMPNN model, can also be seen as a form of regularization, however while the increase in noise leads to improved sequence diversity it also leads to the decrease in amino acid recovery rate. SC regularization, on the other hand, promotes diverse sequences and maintains similar recovery rate.

| Model | Recovery | Change |
|---|---|---|
| GVP
Jing et al. (2020) | 40.2 | – |
| GVP
Hsu et al. (2022) | 42.2 | – |
| GVP + AlphaFold2 data
Hsu et al. (2022) | 38.6 | -3.6 (-8.5%) |
| GVP
(our experiment) | 38.6 | – |
| GVP + SC
(our experiment) | 39.6 | +1.0 (+2.6%) |

Table 6: Comparison of amino acid recovery rate of protein sequences generated by GVP on the test split of CATH dataset. The lighter and darker highlighted blocks represent the results from three experiments: original results from GVP authors, the results from ESM authors, and the results from our experiments. A small difference between the values in first, second and forth rows can be attributed to some discrepancies in experimental settings as well as model initialization. We can see that a simple data augmentation baseline results in 3.6 (or 8.5%) drop of recovery relative to the unaugmented GVP. On the other hand, the use of SC regularization leads to 1.0 (or 2.6%) gain in recovery, signaling the benefit of the proposed distillation approach.

| | Recovery | | Diversity | | Perplexity | |
|---|---|---|---|---|---|---|
| | ProteinMPNN | ProteinMPNN
+SC | ProteinMPNN | ProteinMPNN
+SC | ProteinMPNN | ProteinMPNN
+SC |
| Backbone Noise 0.02 | 43.7 | 43.5 | 22.5 | 24.3 | 5.1 | 5.1 |
| Backbone Noise 0.1 | 39.8 | 40.0 | 28.1 | 30.4 | 5.3 | 5.4 |
| Backbone Noise 0.2 | 36.5 | 36.2 | 31.3 | 34.4 | 5.8 | 5.8 |
| Backbone Noise 0.3 | 33.3 | 33.1 | 33.0 | 37.8 | 6.2 | 6.3 |

Table 7: Evaluation results of ProteinMPNN trained with and without SC regularization (AFDistill trained on TM aug 86K dataset). Structure Consistency (SC) regularization maintains recovery and perplexity rates while improving the diversity of the generated protein sequences. Backbone noise is part of ProteinMPNN model and it can be seen that the increase in noise leads to less accurate amino acid recovery, while it improves sequence diversity. SC regularization, on the other hand, leads to more diverse sequences, while maintaining same recovery rates.

## F  AFDISTILL EVALUATION ON DOWNSTREAM APPLICATIONS

Finally, in Tables 8 9, and 10 we show detailed results for GVP and Graph Transformer inverse folding task as well as protein infilling task. The table combines all the choices for AFDistill pretraining, showing their validation accuracy, and presents the corresponding performance on the downstream application without (top row in each table) and with SC regularization (all the following rows).

| Data | Distill model | | | GVP | | | | |
| | Training | | Validation | Recovery | Diversity | Perplexity | pTM/pLDDT | TM score |
| | Weighted sampling | Focal loss | CE loss | | | | | |
| – | – | – | – | 38.6 | 15.1 | 6.1 | 0.80 | 0.79 |
| . TM 42K | + | – | 1.37 | 36.8 | 22.2 | 6.3 | 0.78 | |
| | + | 1.0 | 1.16 | 37.6 | 21.1 | 6.0 | 0.87 | |
| | + | 3.0 | 1.10 | **39.6** | 21.1 | 5.9 | 0.84 | 0.84 |
| | + | 10.0 | 1.29 | 37.9 | 18.4 | 6.0 | 0.80 | |
| TM augmented 86K | – | – | 2.12 | 38.3 | 22.2 | 5.9 | 0.73 | |
| | + | 1.0 | 2.15 | **39.8** | 22.1 | 5.8 | 0.78 | 0.85 |
| | + | 3.0 | 2.19 | 37.8 | 19.8 | 6.1 | 0.73 | |
| | + | 10.0 | 2.25 | 38.5 | 21.2 | 5.9 | 0.72 | |
| TM synthetic 1M | – | – | 2.90 | 38.8 | 21.4 | 5.8 | 0.73 | |
| | + | 1.0 | 2.55 | **39.1** | 22.5 | 5.9 | 0.77 | 0.81 |
| | + | 3.0 | 2.75 | 39.0 | 21.9 | 5.8 | 0.74 | |
| | + | 10.0 | 3.20 | 39.0 | 22.0 | 5.9 | 0.69 | |
| LDDT 42K | – | – | 3.47 | **39.0** | 18.9 | 5.8 | 0.74 | 0.78 |
| | + | 1.0 | 3.44 | 38.7 | 22.5 | 5.8 | 0.73 | |
| | + | 3.0 | 3.42 | 38.9 | 21.2 | 5.8 | 0.73 | |
| | + | 10.0 | 3.39 | 38.5 | 22.3 | 5.9 | 0.72 | |
| . PLDDT 1M | – | – | 3.27 | **39.3** | 16.5 | 5.9 | 0.76 | 0.79 |
| | + | 1.0 | 3.28 | 38.8 | 15.5 | 5.9 | 0.72 | |
| | + | 3.0 | 3.25 | 38.9 | 18.2 | 5.8 | 0.78 | |
| | + | 10.0 | 3.24 | 38.4 | 16.2 | 6.0 | 0.73 | |
| LDDT chain 42K | – | – | 3.69 | 38.8 | 20.0 | 5.8 | 0.74 | |
| | + | 1.0 | 3.57 | **39.3** | 16.3 | 5.8 | 0.79 | 0.78 |
| | + | 3.0 | 3.63 | 38.9 | 15.9 | 5.9 | 0.72 | |
| | + | 10.0 | 3.59 | 37.9 | 23.2 | 6.0 | 0.73 | |
| pLDDT chain 1M | – | – | 3.29 | 39.4 | 17.4 | 5.8 | 0.78 | |
| | + | 1.0 | 3.36 | 38.7 | 16.3 | 5.8 | 0.76 | |
| | + | 3.0 | 3.30 | **39.6** | 18.3 | 5.7 | 0.79 | 0.77 |
| | + | 10.0 | 3.31 | 38.2 | 20.1 | 6.0 | 0.76 | |
| pLDDT balanced 1M | – | – | 2.63 | 39.1 | 17.1 | 5.8 | 0.75 | 0.82 |
| pLDDT balanced 10M | – | – | 2.43 | 39.3 | 17.7 | 5.9 | 0.73 | |
| pLDDT balanced 60M | – | – | 2.40 | **39.8** | 17.5 | 5.9 | 0.74 | 0.81 |
| pLDDT chain balanced 1M | – | – | 2.45 | 38.6 | 16.6 | 5.9 | 0.73 | |
| pLDDT chain balanced 10M | – | – | 2.24 | 39.1 | 17.8 | 5.8 | 0.73 | |
| pLDDT chain balanced 60M | – | – | 2.21 | **39.7** | 17.9 | 5.9 | 0.74 | 0.82 |

Table 8: Evaluation results of GVP inverse folding task, trained without (top row) and with SC regularization (all other rows). The table combines all the choices for AFDistill pretraining and showing their validation accuracy, as well as the corresponding performance on the downstream application. We select the best performance for each experiment based on the highest recovery rate (marked in bold).

| Data | Distill model | | | Graph Transformer | | | | |
|---|---|---|---|---|---|---|---|---|
| | Training | | Validation | Recovery | Diversity | Perplexity | pTM/pLDDT | TM score |
| | Weighted sampling | Focal loss | CE loss | | | | | |
| – | – | – | – | 25.2 | 72.2 | 7.2 | 0.80 | 0.81 |
| TM 42K | + | – | 1.37 | 24.1 | 74.4 | 7.4 | 0.81 | |
| | + | 1.0 | 1.16 | 25.2 | 73.2 | 7.2 | 0.86 | |
| | + | 3.0 | 1.10 | **25.8** | 74.5 | 7.2 | 0.83 | 0.80 |
| | + | 10.0 | 1.29 | 24.9 | 73.9 | 7.3 | 0.81 | |
| TM augmented 86K | – | – | 2.12 | 25.0 | 73.3 | 7.1 | 0.78 | |
| | + | 1.0 | 2.15 | **25.9** | 74.0 | 7.1 | 0.80 | 0.82 |
| | + | 3.0 | 2.19 | 24.9 | 73.4 | 7.3 | 0.76 | |
| | + | 10.0 | 2.25 | 24.8 | 73.4 | 7.2 | 0.79 | |
| TM synthetic 1M | – | – | 2.90 | 25.3 | 73.2 | 7.1 | 0.72 | |
| | + | 1.0 | 2.55 | **25.5** | 73.9 | 7.2 | 0.79 | 0.75 |
| | + | 3.0 | 2.75 | 25.2 | 73.5 | 7.2 | 0.77 | |
| | + | 10.0 | 3.20 | 24.9 | 74.2 | 7.2 | 0.76 | |
| LDDT 42K | – | – | 3.47 | 25.4 | 73.2 | 7.1 | 0.75 | |
| | + | 1.0 | 3.44 | **25.7** | 74.2 | 7.1 | 0.72 | 0.76 |
| | + | 3.0 | 3.42 | 25.5 | 74.4 | 7.2 | 0.73 | |
| | + | 10.0 | 3.39 | 25.3 | 22.3 | 7.2 | 0.72 | |
| pLDDT 1M | – | – | 3.27 | 25.6 | 73.4 | 7.1 | 0.79 | |
| | + | 1.0 | 3.28 | 25.4 | 74.1 | 7.2 | 0.78 | |
| | + | 3.0 | 3.25 | **25.6** | 74.3 | 7.1 | 0.79 | 0.79 |
| | + | 10.0 | 3.24 | 25.4 | 74.0 | 7.1 | 0.77 | |
| LDDT chain 42K | – | – | 3.69 | 25.3 | 74.1 | 7.2 | 0.76 | |
| | + | 1.0 | 3.57 | **25.8** | 74.3 | 7.1 | 0.75 | 0.80 |
| | + | 3.0 | 3.63 | 25.5 | 74.2 | 7.1 | 0.77 | |
| | + | 10.0 | 3.59 | 25.6 | 74.1 | 7.2 | 0.76 | |
| pLDDT chain 1M | – | – | 3.29 | 25.3 | 74.3 | 7.1 | 0.78 | |
| | + | 1.0 | 3.36 | 25.2 | 74.1 | 7.1 | 0.76 | |
| | + | 3.0 | 3.30 | **25.6** | 74.4 | 7.2 | 0.79 | 0.81 |
| | + | 10.0 | 3.31 | 25.3 | 74.3 | 7.1 | 0.77 | |
| pLDDT balanced 1M | – | – | 2.63 | 25.8 | 74.2 | 7.1 | 0.70 | |
| pLDDT balanced 10M | – | – | 2.43 | 25.7 | 74.5 | 7.1 | 0.73 | |
| pLDDT balanced 60M | – | – | 2.40 | **26.0** | 74.2 | 7.2 | 0.74 | 0.78 |
| pLDDT chain balanced 1M | – | – | 2.45 | 25.7 | 74.3 | 7.1 | 0.72 | |
| pLDDT chain balanced 10M | – | – | 2.24 | 25.9 | 74.4 | 7.1 | 0.74 | |
| pLDDT chain balanced 60M | – | – | 2.21 | **25.9** | 74.5 | 7.2 | 0.72 | 0.77 |

Table 9: Evaluation results of Graph Transformer inverse folding task, trained without (top row) and with SC regularization (all other rows). The table combines all the choices for AFDistill pretraining and showing their validation accuracy, as well as the corresponding performance on the downstream application. We select the best performance for each experiment based on the highest recovery rate (marked in bold).

| Data | Distill model Training | | Validation | CDR Infill Recovery | Diversity | Perplexity | pTM/ pLDDT | TM score |
|------|-------------------|-----------|------------|----------|-----------|------------|----------|----------|
| | Weighted sampling | Focal loss | CE loss | | | | | |
| – | – | – | – | 41.5 | 14.5 | 6.8 | 0.80 | 0.85 |
| TM 42K | + | – | 1.37 | 41.9 | 15.7 | 6.5 | 0.81 | |
| | + | 1.0 | 1.16 | **42.4** | 16.6 | 6.3 | 0.77 | 0.85 |
| | + | 3.0 | 1.10 | 41.7 | 14.6 | 6.7 | 0.78 | |
| | + | 10.0 | 1.29 | 40.8 | 14.4 | 6.6 | 0.79 | |
| TM augmented 86K | – | – | 2.12 | **42.8** | 15.5 | 6.5 | 0.78 | 0.84 |
| | + | 1.0 | 2.15 | 41.6 | 14.8 | 6.6 | 0.74 | |
| | + | 3.0 | 2.19 | 41.3 | 14.6 | 6.7 | 0.76 | |
| | + | 10.0 | 2.25 | 40.9 | 15.4 | 6.8 | 0.79 | |
| TM synthetic 1M | – | – | 2.90 | 41.8 | 16.0 | 6.6 | 0.79 | |
| | + | 1.0 | 2.55 | **41.9** | 15.9 | 6.7 | 0.79 | 0.84 |
| | + | 3.0 | 2.75 | 41.3 | 16.1 | 6.6 | 0.77 | |
| | + | 10.0 | 3.20 | 40.9 | 16.2 | 6.7 | 0.78 | |
| LDDT 42K | – | – | 3.47 | 41.3 | 15.1 | 6.5 | 0.83 | |
| | + | 1.0 | 3.44 | 40.3 | 15.5 | 6.7 | 0.84 | |
| | + | 3.0 | 3.42 | 40.8 | 14.4 | 6.8 | 0.81 | |
| | + | 10.0 | 3.39 | **41.9** | 14.9 | 6.6 | 0.79 | 0.84 |
| pLDDT 1M | – | – | 3.27 | **41.8** | 15.4 | 6.3 | 0.85 | 0.83 |
| | + | 1.0 | 3.28 | 40.7 | 14.3 | 6.5 | 0.85 | |
| | + | 3.0 | 3.25 | 41.7 | 17.2 | 6.5 | 0.84 | |
| | + | 10.0 | 3.24 | 41.6 | 16.1 | 6.6 | 0.85 | |
| LDDT chain 42K | – | – | 3.69 | 40.8 | 15.1 | 6.7 | 0.77 | |
| | + | 1.0 | 3.57 | 40.9 | 15.7 | 6.6 | 0.85 | |
| | + | 3.0 | 3.63 | **41.7** | 15.2 | 6.9 | 0.84 | 0.85 |
| | + | 10.0 | 3.59 | 41.6 | 15.2 | 6.8 | 0.83 | |
| pLDDT chain 1M | – | – | 3.29 | 40.5 | 16.1 | 6.6 | 0.81 | |
| | + | 1.0 | 3.36 | 40.8 | 17.1 | 6.5 | 0.88 | |
| | + | 3.0 | 3.30 | 41.0 | 15.0 | 6.5 | 0.85 | |
| | + | 10.0 | 3.31 | **41.8** | 15.4 | 6.3 | 0.87 | 0.85 |
| pLDDT balanced 1M | – | – | 2.63 | **42.1** | 15.8 | 6.4 | 0.75 | 0.83 |
| pLDDT balanced 10M | – | – | 2.43 | 42.0 | 14.9 | 7.0 | 0.76 | |
| pLDDT balanced 60M | – | – | 2.40 | 42.1 | 16.5 | 6.3 | 0.73 | |
| pLDDT chain balanced 1M | – | – | 2.45 | 41.1 | 18.0 | 6.1 | 0.75 | |
| pLDDT chain balanced 10M | – | – | 2.24 | 41.3 | 17.0 | 6.7 | 0.74 | |
| pLDDT chain balanced 60M | – | – | 2.21 | **41.9** | 17.5 | 6.3 | 0.73 | 0.83 |

Table 10: Evaluation results of Protein Infilling task, trained without (top row) and with SC regularization (all other rows). The table combines all the choices for AFDistill pretraining and showing their validation accuracy, as well as the corresponding performance on the downstream application. We select the best performance for each experiment based on the highest recovery rate (marked in bold).

