# OpenReview forum: "AlphaFold Distillation for Improved Inverse Protein Folding"
_ICLR.cc/2023/Conference — Submitted to ICLR 2023_

### Official Review · Reviewer_Y6Az · 2022-10-24

**Confidence:** 3
**Correctness:** 3
**Technical Novelty And Significance:** 2
**Empirical Novelty And Significance:** 3
**Recommendation:** 6

**Clarity, Quality, Novelty And Reproducibility:**

Clarity: The paper is kind of hard to follow for general machine learning researchers. Considering this is machine learning conference, I suggest authors to significantly improve the presentation to make it more friendly to general ML audience. See details below.
1. The main paper starts by discussing how to distill AlphaFold model. The audience of the ICLR conference is mostly machine learning researchers, who may not be familiar with proteins. I’d like to suggest that the authors add a preliminary section to introduce the basic concepts of proteins, the problem definition, and high-level ideas of related methods.
2. The authors mix the method section with the experimental setup and results. This makes the paper more like a technical report that requires readers’ knowledge of the task and datasets.
3. There are 10 figures and 2 tables in the main paper. The readers need to switch between different pages for finding results with similar meanings. This issue can be solved by better illustration. For example, the contents of Fig. 7,9(,10) can be organized in one table.

Quality is pretty high with extensive experiments and detailed methodology.

For novelty, I feel like the paper's idea is not surprising to me. The idea of distilling Alphafold into inverse folding is somewhat new, but the ML components like the distillation method and network implementation are all established methods.

The author promised to provide the source code upon acceptance. I suppose reproducibility should not be a problem.

**Strength And Weaknesses:**

Strength:
1. Address an important problem with a well-motivated method.
2. Simple techniques with impressive results, especially in protein diversity.

Weakness:
1. Lack of technical novelty. The models and distillation methods are adapted from existing works without non-trivial modifications.
2. Lack of related work on knowledge distillation.
3. The presentation is not friendly to readers that are unfamiliar with inverse protein design.

**Summary Of The Paper:**

The paper focuses on inverse protein folding, a fundamental and challenging problem in protein science. The authors propose a new regularization term by including the trained AlphaFold model and its predicted structural confidence metric. To improve the efficiency of the model optimization loop, they distill the AlphaFold into a smaller model. They perform extensive experiments with different protein design models to show generality of the regularization trick. Experimental results show that the model with regularization achieves better sequence recovery rate and large improvement in protein diversity, while preserving the structural consistency.

**Summary Of The Review:**

A method distilling Alphafold for inverse folding, which is an important topic for computational protein design. The idea of distilling the advanced protein structure prediction model for inverse folding might be interesting to the specific "ML for protein" audience, while generally the ML models and methods are all just existing works. Experiments are sound and comprehensive.

Questions:
1. It’s interesting to see that such a regularization term can increase sequence diversity. Could the authors provide some explanations and in-depth analysis of this phenomenon?

Minor points:
1. In the caption of Fig.3. Line 5: TM/LDTT -> TM/LDDT
2. In Fig. 7,9,10. The cyan bar is overlapped with the purple bar, which makes the color similar to the blue bar. Since the baseline value is the same for all methods in one figure, I suggest using a dashed red line rather than a cyan bar.

---

> ### Author Response · Authors · 2022-11-14
> **Response**
>
> Thank you for your overall positive assessment of our work. We are sorry that some parts of our work were hard to follow and we'll try to improve the writing to make it accessible to a wider audience.
>
> 1. Lack of technical novelty.
>
> It is true that we used the existing distillation approach to build AFDistill model. However, our main contribution and novelty lies in the computational protein design domain where we proposed to bypass protein structure estimation in AlphaFold2 and directly estimate pTM/pLDDT scores from a given protein sequence. These scores are then used as a training regularization in protein design or infilling problems.
>
> 2. Lack of related work on knowledge distillation.
>
> Thank you for pointing this out. We'll include relevant related work.
>
> 3. Not friendly to unfamiliar readers.
>
> We are sorry about that and try to expand the paper to include more definitions and citations to help unfamiliar reader to clarify some common terms related to inverse protein design. Please see Section A in the Appendix for the added material.
>
> 4. Many figures.
>
> We'll try to reorganize the paper to be easier to read. Thanks for the recommendations.
>
> 5. Regularization increasing diversity.
>
> The intuition behind this phenomenon can explained as follows. For a given protein sequence AFDistill computes pTM/pLDDT that estimates the structural consistency score (SC), i.e., estimates the closeness between ground truth and estimated protein structures. However, since AFDistill does not have access to any of the structures (neither estimated not ground truth), for a given protein sequence, AFDistill can only estimate pTM/pLDDT based on what it has seen during training. For example, a degenerate protein sequence would generally result in a poorly estimated structure, and consequently, low pTM/pLDDT scores. While a well-formed protein sequences usually lead to accurate structure estimates with high pTM/pLDDT scores.
>
> In our experiments for protein design (e.g., GVP) the objective is CE + SC (cross-entropy  + AFDistill structure consistency score), and we observed that without CE loss while keeping SC, results in degenerated protein sequences. This is expected, since AFDistill alone cannot guide GVP which sequence it should generate to match the given input 3D structure. (AFDistill has no information about the structure, and since many of the relevant sequences can have high pTM/pLDDT, all of them could be good candidates, and this promotes diversity.) On the other hand, when only the CE is present, it guides GVP to recover the specific ground truth sequence for a given 3D structure, and this promotes accuracy. Consequently, when both CE and SC terms are present, we get a full benefit - accurate recovery and diversity at the same time. We'll include some experiments to illustrate this discussion. Please see Section D.2 in the Appendix.

---

### Official Review · Reviewer_jZLG · 2022-10-26

**Confidence:** 2
**Correctness:** 3
**Technical Novelty And Significance:** 2
**Empirical Novelty And Significance:** 2
**Recommendation:** 3

**Clarity, Quality, Novelty And Reproducibility:**

**Minor Questions/Problems:**

Section 2: Jumper et al’s work should be named AlphaFold2, instead of AlphaFold.

Section 3: “Traditionally, the distillation would be done using soft labels…”. reference(s) is missing to support ‘traditionally’.

Section 3: “we do not use the probabilities as they are harder to collect…”. it is unclear that the probabilities are harder than what to collect?

Section 3: “hard labels…computed based on AlphaFold’s predicted 3D structure”. It is very confusing: in the previous line a hard label refers to **ground truth** classes, and it is not clear why AlphaFold’s **predicted** 3D structures are considered ‘hard labels’.

Figure 2: what is the meaning of ‘Logits’ in the upper purple box? Does it refer to a softmax activation function? Or a Logistic regression?

Section 3.3: “vocabulary size to 50, corresponding to discretizing…”. What is the relationship between the vocabulary size and pTM/pLDDT range? (as the authors used the word ‘corresponding to’)

**Strength And Weaknesses:**

**Strengths:**

1. The paper overall is easy to follow, and the model construction is clearly introduced.

2. According to the experimental results reported in the main paper, the main claimed contributions of “efficiently interpreting the protein sequence” and “appliable to other tasks (protein infilling)” are supported with evidence (although this evidence is flawed, see weaknesses).

**Weaknesses:**

1. The system design can be better motivated. In particular, why it is essential to infer protein structure from the predicted sequence for an inverse folding problem? Does it follow some convention or some specific empirical meaning?

2. It is questionable whether the comparison of 'the efficiency' of the proposed method against baselines is fundamentally fair. It seems the only (in)direct comparison is in the left plot of figure 2, where the inference time is compared against different folding models. However, if the reviewer understands correctly, the distillation model itself requires running AlphaFold2 to obtain the predicted labels (which are later used for training). In other words, Figure 2 actually compares a fraction of the proposed model with the entire competitor model.

3. The comparison of computational efficiency is incomplete. While the authors report the inference speed with a chain of 500 amino acids, it is not clear if the proposed method also performs swift inference speed in large-scale protein sequences. As the chain length can go beyond thousands of amino acids, the practical importance of the proposed model is questionable. While the authors claimed “truncated proteins up to length 500 to reduce computational complexity” (above Figure 4), other language models (such as ESM-IF1) work well on amino acid chains of up to 1024 tokens.

4. More experimental results should be included (in the main text) to support the merit of the proposed method. For instance, as the designed module mainly solves the inverse folding problem, at least some specific baselines should be compared, such as ESM-1v and ESM-IF1. Meanwhile, the designed computational blocks should be investigated with detailed ablation studies.

**Summary Of The Paper:**

This work proposed a knowledge distillation module for protein inverse folding, i.e., generating a protein sequence by its 3D structure. The proposed system recovers more diverse amino acid chains with a lower perplexity and higher recovery rate efficiently.





**Summary Of The Review:**

The submission is not sufficient to be published in ICLR.

---

> ### Author Response · Authors · 2022-11-14
> **Response**
>
> Thank you for your review and the comments.
>
> 1. Why infer structure of the generated protein.
>
> The motivation is similar to "cycle"-type works (e.g., cycle-GAN). For inverse protein design, given a 3D structure, the objective is to generate a protein sequence consistent with that structure. Traditionally, cross-entropy (CE) loss is used to compare generated and ground truth sequence. However, it is known that a given 3D structure can have many corresponding protein sequences that fold into that structure. Therefore, simply forcing the generated sequence to be the same as ground truth sequence is very limiting. For this purpose, we proposed AFDistill that takes a generated protein and outputs its structure consistency score (pTM or pLDDT) (which roughly answers the question: how close is the estimated protein structure of a given generated sequence with respect to the ground truth structure?). For example, if pTM is high (good match between structures) while CE loss is high (poor sequence match), this might still correspond to a good generated protein sequence that should not be discarded or penalized.
>
> 2. Inference time in Figure 2.
>
> The timings in Figure 2 are the actual inference times for each of the method given a protein sequence of length 500. The inference of AFDistill model does not involve any running of AlphaFold2. The only time where AlphaFold2 is involved (implicitly) is in generating sequence-score pair set for AFDistil training (but we don't even run AlphaFold2 since all these data are available for download from repository).
>
> Moreover, recall that AFDistill is a ProtBert model with modified classification head that outputs pTM or pLDDT score. It makes AFDistill fast and fully differentiable. Also note that AFDistill does not estimate any protein structures, as opposed to all other methods in Figure 2, which do estimate the structures, making them much slower and impractical for in-loop optimization. In fact, the main reason we decided to do distillation of pTM/pLDDT scores is to bypass expensive structure estimation and directly estimate pTM/pLDDT, otherwise we could  have directly included AlphaFold2 in the optimization loop (assuming this would still be differentiable). Please see Figure 1 illustrating this.
>
> 3. Chain length 500.
>
> Truncation is a common practice when the computational resources are limited. When using sequences of length up to 500, capturing single chain proteins, during AFDistill training, we had to limit batch size to 10. Increasing to longer protein sequences would make batch size even smaller and require additional resources to maintain good training. However, clearly AFDistill can handle any sequence length, provided enough GPU memory. In fact, we updated Figure 2 with additional results (we made a typo and the corrected timing for AFDistill is 0.02 seconds): sequence length 1024 - inference time 0.028 seconds, sequence length 2048 - inference time 0.035 seconds. The key point to observe is that AFDistill bypasses protein structure estimation and directly outputs pTM/pLDDT score, while AlphaFold2 and similar methods predict the structure, making them necessarily slower approaches, irrespective of the sequence length.
>
> 4. More experiments.
>
> Thank you for the suggestions. We agree that the comparison with ESM-IF indeed should be interesting - comparing GVP + AFDistill vs GVP + augmented data. In fact, Table 1 in ESM-IF paper compares the original GVP with the GVP trained on augmented dataset (CATH + AF2-generated structure/sequence pairs). The simple augmented baseline showed worse performance than the original GVP, and the authors had to make GVP much larger to get any benefit from the data augmentation. Please refer to Section E.1 and E.2 in Appendix for newly added results.
>
> Could the reviewer please clarify the part on "Meanwhile, the designed computational blocks should be investigated with detailed ablation studies." Which computational blocks they would like to see ablated?
>
> 5. Minor questions/comments.
>
> Thank you for pointing these out. Clarification: using AlphaFold2 dataset, for each protein sequence we collect pTM/pLDDT scores (estimated by AlphaFold2), as well as TM/LDDT scores (which we computed, using ground truth and AF2 estimated structures).
>
> Logits in Figure 3 refer to vectors before entering the softmax activation function.
>
> pTM/pLDDT values have range (0,1) and we discretize this range into 50 bins.

---

### Official Review · Reviewer_LQja · 2022-10-31

**Confidence:** 4
**Clarity, Quality, Novelty And Reproducibility:** The paper is clearly written and the …
**Correctness:** 4
**Technical Novelty And Significance:** 2
**Empirical Novelty And Significance:** 4
**Recommendation:** 8

**Strength And Weaknesses:**

+ The main idea is a simplistic distillation of the AlphaFold model to be used as a consistency regularization for other protein modeling tasks.
+ The papers shows that such as simple idea works in drastically improving efficiency of the folding model and the cosistency regulariztion is clearly effective for multiple tasks.
+ The correlation of the distilled model's scores seem to be acceptably high with respect to the original AF's scores.
+ The general performance on protein design seems to significantly improve over the recent baselines in terms of ground-truth sequence likelihood and amino-acids recovery.
+ Different types of inverse folding backbones is used in conjunction with the consistency loss to demonstrate the universality of its usefulness.
+ Interestingly, the diversity of the obtained well-structured sequences is improved over the baselines.
+ Results on (structure-preserving) sequence infilling is also encouraging.


Some questions:
- how does the distilled model perform if trained from scratch and not from a pre-trained language model?
- can the fact that the model is pre-trained as a self-supervised language model suggest that it might have memorized some sequences that it has to generate on the validation set?


**Summary Of The Paper:**

The paper mainly tackles the problem of protein design through training of an inverse folding model. The procedure involves using a protein folding model such as AlphaFold as a consistency regularizer along with the sequence generation task. To make such learning feasible in terms of computational efficiency and training time, it distills AlphaFold folding model into a smaller, more efficient, but similarly-accurate BERT-like transformer model.

**Summary Of The Review:**

The paper proposes a simple consistency regularization method for protein modeling, with the most focus on protein design. Such a simple setup achieves encouraging results which significantly improve over recent baselines on important tasks and aspects. Conditioned on a satisfying discussion around my questions above I find the paper a clearly well-performed application paper on an important topic of relatively-wide interest.

-------- post rebuttal ---------
I read the other reviews and the authors' response. I believe the paper has clear merits for publication as the approach is reasonable and novel, the diversity of baselines are increased with the new SC mechanism while mostly maintaining the recovery rate, and the efficiency of distilling model for AF2 scores.

---

> ### Author Response · Authors · 2022-11-14
> **Response**
>
> Thank you for the appreciation of the main ideas of our work and the positive evaluation of the results.
>
> 1. AFDistill from scratch.
>
> Yes, we trained few models from scratch, but observed worse performance. As an example, we trained AFDistill from scratch on TM42K dataset. The validation CE loss during distillation was 1.5 (versus 1.1 when using pre-trained ProtBert model). Moreover, training of AFDistill model from scratch takes longer (3 days vs 1 day). When regularizing GVP with AFDistill from scratch, we get similar recovery rate (39.4 vs 39.6) but lower sequence diversity (15.9 vs 21.1).  Please refer to Section D.1 in the Appendix.
>
> 2. Sequence memorization in the pre-trained ProtBert.
>
> This is an interesting question. The pre-training of ProtBert follows traditional masked token prediction, while AFDistill, based on pre-trained ProtBert, has entirely different objective of predicting pTM or pLDDT. During regularization (e.g., in GVP training), the AFDistill model receives GVP-generated protein sequences which are unlikely to be similar to what ProtBert has seen during training. Moreover, during AFDistill training we ensured that the training data has no overlap with the test set of CATH dataset. Finally, please note that AFDistill is not a generative model and does not generate protein sequences, it is a scorer, that receives a protein sequence as input and outputs the estimated pTM or pLDDT score.
>
> In earlier stages of GVP training these sequences are mostly degenerate (e.g, repetition of a single amino acid AAAAAA), while in later stages the sequences are well-formed but quite diverse, therefore it is unlikely that AFDistill has seen these exact sequences.

---

### Official Review · Reviewer_QCkw · 2022-11-01

**Confidence:** 4
**Clarity, Quality, Novelty And Reproducibility:** n/a
**Correctness:** 2
**Technical Novelty And Significance:** 2
**Empirical Novelty And Significance:** 2
**Recommendation:** 3

**Strength And Weaknesses:**

Overall, the motivation for incorporating folding models to improve the structural fidelity of inverse folding models is straightforward, and resorting to knowledge distillation makes sense. However, my major concerns lie in both the model and evaluation parts. Please refer to the questions for more details.

Questions:

1. The baseline models, GVP, are weak. It remains unclear whether the proposed idea could bring consistent improvements when strong inverse folding models are considered, such as protein mpnn?
2.  Since additional 907k AF2 predicted proteins are used to build the distilled model, a simple baseline in which all of these predicted data are directly used for inverse folding training should be carefully considered and compared. What’s more, does the proposed approach, which requires significant additional overhead in terms of both distillation and regularized training, complement with a simple data augmentation approach? This important question remains unclear.
3. Comparisons to the state of the arts methods are missing, e.g., ESM-IF (Hsu+ ICML 2022).


**Summary Of The Paper:**

This paper tries to incorporate forward folding/structure prediction models, i.g., AlphaFold 2 in particular, to provide informative feedback for inverse folding models during training. One of the challenges here is forward folding models are usually too large and computationally expensive to be integrated into the learning framework. To tackle this, this paper proposes to distill AF2 by a ProtBert, a Transformer based language model, in terms of AF2’s confidence metrics, i.e., pLDDT and pTM scores. After distillation, the distilled score model is used to provide signals regarding structural consistency for learning inverse folding models. Experiments on fixed backbone design (CATH 4.2) and antibody CDR infilling (SabDab) show the effectiveness of the proposed approach.



**Summary Of The Review:**

-

---

> ### Author Response · Authors · 2022-11-14
> **Response**
>
> Thank you for the provided review and comments.
>
> 1. GVP is weak. Add MPNN.
>
> Thank you for this recommendation. We have not included them due to limited computational resources and space. However, following your suggestion, we updated the paper with new results.  Please refer to Section E.2 in the Appendix.
>
> Related to GVP, it is true that the model has lower amino acid recovery rate. Moreover, we observed that AFDistill tends to have smaller impact on the recovery improvement but has much larger impact on increasing the sequence diversity. Therefore regularizing GVP training with AFDistill led to a significantly more diverse sequences while maintaining similar recovery rate. Therefore, we argue that it is misleading to directly compare GVP+AFDistil with other approaches, including MPNN or ESM-IF. ***Rather, the effect of proposed AFDistill regularization on GVP needs to be compared with the original GVP. Similarly, MPNN+AFDistill needs to be compared with the original MPNN to evaluate the effect.***
>
> Finally, please note that the ***goal of our paper is not to create a method with the highest amino acid recovery rate. In fact, as we stated in the Introduction, chasing high recovery rate in itself is the wrong objective*** for computational protein design. For example, what would be the value of the system having close to 100% recovery rate? This would result in re-creating already known protein sequences with little novelty and no practical value. We argue that ***the combination of good recovery (>30%), high structural consistency,  and high diversity rates should be the objective for inverse protein design***. In this case, the designed sequences exhibit novel and diverse functionalities while maintaining the consistent fold structure.
>
> 2. Simple data augmentation baseline.
>
> The suggested baseline method is exactly one of the baselines in ESM-IF paper. Specifically, in Table 1 of that paper the authors compare the original GVP with the GVP trained on augmented dataset (CATH + AF2-generated structure/sequence pairs). Interestingly, the simple augmented baseline showed worse performance than the original GVP, and the authors had to make GVP much larger to get any benefit from the data augmentation.
>
> On the other hand, our AFDistill regularization was able to improve the original GVP (with modest recovery and significant diversity gains). Moreover, the distillation overhead is amortized - train once use everywhere. We paid extra cost to train AFDistill but then applied it in many downstream tasks with little overhead. Data augmentation would require additional significant cost in every application. Finally, AFDistill offers unique advantage - it maintains recovery and structural consistency of the original model while introducing diversity into the generated sequences.
>
> 3. Compare with ESM-IF.
>
> Thank you for the suggestion, we'll add this to the paper. It is indeed meaningful to discuss and compare AFDistill regularization with the data augmentation as applied to the original GVP (1M) in that paper. However, GVP-large (21M) and GVP-Transformer (142M) are the models of entirely different scale and not directly comparable to GVP+AFDistill. Please refer to Section E.1 in the Appendix.

---

### Author Response · Authors · 2022-11-14
**Revised paper**

Based on the comments from reviewers, we have revised the paper. Here is the summary of the changes (we used red color to mark them in the paper):
- Updated Figure 2 and its caption
- Section A in Appendix covers some background on Protein Design
- Section D.1 in Appendix discusses the effects of using AFDistill when trained from scratch
- Section D.2 in Appendix covers the effect of structure consistency (SC) score on GVP performance
- Section E.1 in Appendix compares AFDistill applied to GVP to ESM-IF
- Section E.2 in Appendix compares GVP + AFDistill to ProteinMPNN (to be completed soon)

---

### Author Response · Authors · 2022-11-16
**Revised paper**

We uploaded another updated version of the paper which contains new results on ProteinMPNN experiments (see Section E.2 in the Appendix).

---

### Author Response · Authors · 2022-11-18
**Reviewers**

Dear Reviewers,

Thank you for your review comments. Considering the **deadline of Discussion Stage 1** is approaching, is there anything else we can do for our paper?

Thank you,

Authors.

---

### Decision · Program_Chairs · 2023-01-20

**Decision:**

Reject

**Justification For Why Not Higher Score:**

While the authors should be commended by working to improve their paper and revise during the discussion, the reviewers still maintain main concern for this work that would be ideally addressed (as detailed in the AC_reviewer meeting notes above) before acceptance.

**Justification For Why Not Lower Score:**

N/A

**Metareview: Summary, Strengths And Weaknesses:**

This work proposed a distillation module for protein inverse folding, an important problem in protein design and engineering using machine learning approach. The author showed that their method could recover a set of sequences that show reasonably good performance as measured by the metrics of recovery, structure consistency, and diversity. The overall presentation is clear and through the revision the authors also showed good amount of improvement. The main concern is the improvement over state-of-the-art as well as the demonstration/comparison of the methods, given recent advancement in this area, is overall limited.

**Summary Of Ac-Reviewer Meeting:**

The AC and three reviewers were able to meet to discuss this paper (the meeting was rescheduled to try to accommodate everyone's schedule, but one reviewer was not available). Below is a summary of major points raised by the reviewers. All reviewers at the meeting commended the authors' work on improving the manuscript but all still feel strongly that the paper should be enhanced (as detailed below) before it is suitable for acceptance, so the recommendation was made accordingly.

- major baseline concern: comparison is not quite strong given recent advancement in the inverse protein folding field, such as the ESM-IF that was mentioned by several reviewers.
- structural consistency is a useful metric but overall the reviewers thought it is not a quite fair metric to emphasize.
- the authors have not yet demonstrate the scalability of the model/method, thus, the question remains that would this approach holds the advantage when applied in larger scale, which the reviewers thought is an important point in the context of the progress in this field.
- the machine learning innovation is limited, so the overall scope of this work would not raise significant excitement for the topics of this conference, but would be potentially a good submission for an alternative avenue with more focus on the application of machine learning in protein design and protein folding.